# Aspartate β-Hydroxylase Is Upregulated in Head and Neck Squamous Cell Carcinoma and Regulates Invasiveness in Cancer Cell Models

**DOI:** 10.3390/ijms25094998

**Published:** 2024-05-03

**Authors:** Pritha Mukherjee, Xin Zhou, Susana Galli, Bruce Davidson, Lihua Zhang, Jaeil Ahn, Reem Aljuhani, Julius Benicky, Laurie Ailles, Vitor H. Pomin, Mark Olsen, Radoslav Goldman

**Affiliations:** 1Department of Oncology, Lombardi Comprehensive Cancer Center, Georgetown University, Washington, DC 20057, USA; 2Clinical and Translational Glycoscience Research Center, Georgetown University, Washington, DC 20057, USA; 3Biotechnology Program, Northern Virginia Community College, Manassas, VA 20109, USA; 4Department of Biochemistry and Molecular & Cellular Biology, Georgetown University, Washington, DC 20057, USA; 5Department of Otolaryngology-Head and Neck Surgery, MedStar Georgetown University Hospital, Washington, DC 20057, USA; 6Department of Biostatistics, Bioinformatics and Biomathematics, Georgetown University, Washington, DC 20057, USA; 7Department of Medical Biophysics, University of Toronto, Toronto, ON M5G 1L7, Canada; 8Department of BioMolecular Sciences, University of Mississippi, Oxford, MS 38677, USA; vpomin@olemiss.edu; 9Research Institute of Pharmaceutical Sciences, School of Pharmacy, University of Mississippi, Oxford, MS 38677, USA; 10Department of Pharmaceutical Sciences, College of Pharmacy Glendale Campus, Midwestern University, Glendale, AZ 85308, USA; 11Pharmacometrics Center of Excellence, Midwestern University, Downers Grove, IL 60515, USA

**Keywords:** HNSCC, aspartate β-hydroxylase (ASPH), heparan 6-O-endosulfatase 2 (SULF2), MO-I-1151, HfFucCS, primary head and neck CAF, SCC35 co-culture spheroid

## Abstract

Aspartate β-hydroxylase (ASPH) is a protein associated with malignancy in a wide range of tumors. We hypothesize that inhibition of ASPH activity could have anti-tumor properties in patients with head and neck cancer. In this study, we screened tumor tissues of 155 head and neck squamous cell carcinoma (HNSCC) patients for the expression of ASPH using immunohistochemistry. We used an ASPH inhibitor, MO-I-1151, known to inhibit the catalytic activity of ASPH in the endoplasmic reticulum, to show its inhibitory effect on the migration of SCC35 head and neck cancer cells in cell monolayers and in matrix-embedded spheroid co-cultures with primary cancer-associated fibroblast (CAF) CAF 61137 of head and neck origin. We also studied a combined effect of MO-I-1151 and HfFucCS, an inhibitor of invasion-blocking heparan 6-O-endosulfatase activity. We found ASPH was upregulated in HNSCC tumors compared to the adjacent normal tissues. ASPH was uniformly high in expression, irrespective of tumor stage. High expression of ASPH in tumors led us to consider it as a therapeutic target in cell line models. ASPH inhibitor MO-I-1151 had significant effects on reducing migration and invasion of head and neck cancer cells, both in monolayers and matrix-embedded spheroids. The combination of the two enzyme inhibitors showed an additive effect on restricting invasion in the HNSCC cell monolayers and in the CAF-containing co-culture spheroids. We identify ASPH as an abundant protein in HNSCC tumors. Targeting ASPH with inhibitor MO-I-1151 effectively reduces CAF-mediated cellular invasion in cancer cell models. We propose that the additive effect of MO-I-1151 with HfFucCS, an inhibitor of heparan 6-O-endosulfatases, on HNSCC cells could improve interventions and needs to be further explored.

## 1. Introduction

Aspartate β-hydroxylase (ASPH) belongs to the alpha-ketoglutarate-dependent dioxygenase family of proteins. It has been identified as one of the cell surface proteins associated with malignant transformation of 70–90% of tumors, mainly in the cells of breast, hepatic, colon, pancreatic, and neural origin [1,2,3,4,5,6,7,8,9]. ASPH catalyzes hydroxylation of aspartic acid or asparagine residues in EGF-like domains of several proteins, including clotting factors, extracellular matrix proteins, low-density lipoprotein receptor, Notch homologues, and Notch ligand homologues [5,9,10,11]. It is a type II transmembrane protein with a highly conserved sequence and, besides cell-surface localization, it has been also detected in the endoplasmic and sarcoplasmic reticula. Recent reports describe its presence in mitochondria in specific cancer types [5]. The mature ASPH is transported from the endoplasmic reticulum to the plasma membrane, which exposes the C-terminal region to the extracellular environment. The enzymatic activity of cell surface ASPH has been associated with enhanced cell motility, migration, and invasion, metastatic spread, and drug resistance in solid tumors through its impact on a Notch, proto-oncogene tyrosine-protein kinase Src, phosphoinositide 3-kinases (PI3Ks), and mitogen-activated protein kinase (MAPK) signaling pathways [4,5,11,12].

Specific and selective small molecule inhibitors (SMIs) have been designed to target the hydroxylase activity of ASPH; these compounds inhibit tumor development and metastasis [1,3,5,13]. Antibody-drug conjugate systems, bio-nanoparticle-based therapeutic vaccines, and dendritic cells fused to the ASPH protein have yielded substantial antitumor effects in cell lines and animal models [14,15,16,17]. We and others have developed SMIs of ASPH based on the crystal structure of the ASPH catalytic site. The first ASPH SMIs were the tetronimides MO-I-500 and MO-I-1100 [18,19] followed by MO-I-1151 and MO-I-1182 [3,5], which are modified with trifluoromethyl and carboxymethyl groups, respectively. MO-I-1151 and MO-I-1182 show enhanced activity [3,20]; they not only inhibit catalytic activity but also penetrate cells [5,21]. MO-I-1100 prevented cell migration, invasion and metastasis in hepatocellular carcinoma by modulating the Notch pathway [1]. MO-I-1144 inhibited tumor development and metastasis in colorectal cancer by decreasing Notch expression [22].

ASPH is a documented oncogene in several cancers [5,11] but has not been studied in head and neck squamous cell carcinoma to our knowledge. Besides the inhibition of colorectal cancer [22], ASPH inhibition blocked cell invasion, EMT, and metastasis in pancreatic and hepatocellular carcinomas by interacting with vimentin [2,12]. In pancreatic cancer, ASPH has also been reported to interact with ADAM 12/15, thus activating SRC kinase pathway proteins that control MMP-mediated extracellular matrix degradation and tumor invasion [23]. Tumor cell invasion into the local tissue initiates a metastatic cascade important for the prognosis of cancer patients [24,25]. In the early part of the metastatic process, the tumor cells acquire the ability to penetrate the basement membrane and ECM. The cell–matrix and cell–cell interactions enable malignant tumor cells to invade the surrounding stroma [26,27,28]. Cancer-associated fibroblasts (CAF) are an important mediator of cancer cell invasiveness [29,30] and could contribute to the ASPH function in HNSCC. Single-cell transcriptomic analysis of HNSCC shows that ASPH is expressed in cancer cells but also in the cancer-associated fibroblasts (CAFs) and endothelial cells [31]. In addition, CAFs effectively deposit and remodel the ECM in the tumor micro-environment (TME) [32], and stromal alpha-SMA is an independent prognostic factor in OSCC patients [33]. In this study, we therefore examine the expression of ASPH in HNSCC tumors, and we use a CAF-cancer cell co-culture spheroid model [34] to evaluate the impact of ASPH inhibition on cancer cell invasion.

In summary, we examined the expression of ASPH in tissues of 155 HNSCC patients and in cell models of HNSCC invasion. We tested the ability of the MO-I-1151 ASPH inhibitor alone or in combination with a recently described SULF2 inhibitor HfFucCS [34] with regard to the invasion of HNSCC tumor cells in a spheroid co-culture model with a primary HNSCC cancer-associated fibroblast (CAF 61137). Our results show that inhibition of ASPH inhibits invasion of the HNSCC cell line into Matrigel and that the effect is additive with the SULF2 inhibitor HfFucCS. The results warrant further mechanistic and in vivo studies of ASPH in HNSCC.

## 2. Results

### 2.1. Expression of ASPH in HNSCC Tissues

ASPH staining of 142 patients with primary HNSCC tumors, 10 patients with lymph node metastasis, and 3 patients with CIS were evaluated on our TMA (Figure 1). A subset of the patients (n = 35) had adjacent normal cores represented on the TMA in addition to the primary tumor (Figure 1A). We present higher magnification images of tissue sections showing ASPH expression and distribution in the tumor and adjacent normal tissues in Appendix A. The mean ASPH score (Intensity + Distribution) in primary tumors (mean = 5.0, SD = 0.40) is significantly higher (*p* < 0.0001) than the mean score in the adjacent normal tissues (mean = 3.2, SD = 0.98) (Figure 1B). Analysis of the TCGA and CPTAC dataset for 43 and 64 HNSCC patients, respectively, corroborated our data, showing that ASPH is significantly higher in tumors compared to paired normal tissues at both mRNA (*p* = 0.0004) and protein (*p* = 0.0026) expression levels (Appendix A). The lymph node metastasis showed high expression of ASPH (mean = 5.0, SD = 0.50) similar to the primary tumors; the node scores were also significantly higher (*p* < 0.0001) compared to the normal tissue cores (Figure 1B). The CIS cores stain positive (mean = 5.0), which suggests that ASPH is elevated early, already at the pre-cancerous stage, and is uniformly high at all stages of the HNSCC tumors. However, the CIS cases are poorly represented on our TMA (n = 3), and further studies will need to confirm this result (Figure 1B). Patients with paired primary tumor and lymph node cores (n = 17) showed no difference in mean score (Figure 1C). The uniformly high expression of ASPH across all tumor stages prevented a meaningful analysis of the impact of ASPH on patient survival in our cohort of samples. In our patient cohort, we notice that the intensity of ASPH expression correlated with the grade of primary tumors. The mean scores increase from 1.9 (w), to 2.0 (m), and 2.2 (*p*), which shows a significant increase in poorly differentiated tumors (*p* < 0.05 for ‘w’ to ‘m’, and *p* < 0.001 for ‘m’ to ‘p’ grades), expected to be more aggressive. ASPH expression does not differ significantly between the tumor epithelium and stroma either in intensity or in distribution, which shows that CAF and other non-cancer cells are an important source of ASPH.

### 2.2. MO-1151 Inhibits Migration of HNSCC Cells

We quantified the migration of SCC35 cells in adherent cultures using a wound healing assay. Our results show that MO-I-1151 reduced the migration of cells and delayed wound closure compared to the untreated controls (Figure 2A). Two doses of the inhibitor were tested, and both doses significantly reduced the wound closure at 24 h of treatment, MO-I-1151_Low_ (5µM) 54% (*p* < 0.0001) and MO-I-1151_High_ (25 µM) 52% (*p* < 0.0001) (Figure 2B). The lower dose was sufficient to arrest migration of the SCC35 cells. The untreated controls closed the wound by 48 h in contrast to the treated cells at both doses.

### 2.3. MO-1151 Inhibits CAF-Mediated Invasion of Spheroids in Matrigel

We used a Matrigel-embedded co-culture spheroid model of SCC35 tumor cells with CAF 61137, optimized previously [34], to evaluate the impact of MO-I-1151 on cell invasion (Figure 3A). The CAF-supported tumor cell invasion was recorded on day 5, and our results show that MO-I-1151 reduced the area of the co-culture spheroids by 44% (*p* < 0.0001) for MO-I-1151_Low_ and 45% (*p* < 0.0001) for MO-I-1151_High_ (Figure 3B) compared to the untreated controls. Inverse circularity of the treated spheroids was also significantly reduced, by 61% at the MO-I-1151_Low_ (*p* < 0.0001) and 70% (*p* < 0.0001) at the MO-I-1151_high_ doses (Figure 3C). These results show that MO-I-1151 decreases cancer cell invasion at the CAF–cancer cell interface.

### 2.4. Combination of MO-I-1151 with HfFucCS Has an Additive Effect on Migration of HNSCC Cells

We have shown previously that the marine fucosylated chondroitin sulfate HfFucCS, a newly identified inhibitor of heparan 6-O-endosulfatase SULF2, is an inhibitor of HNSCC tumor cell invasion [34]. We treated the SCC35 cells with a combination of the MO-I-1151 and HfFucCS inhibitors to see if they together inhibit the invasion more efficiently (Figure 4A). We observed that, compared to the untreated controls, the migration of SCC35 cells in adherent cultures was reduced by 74% for the HfFucCS + MO-I-1151_Low_ and 91% for the HfFucCS + MO-I-1151_High_ treatment groups (Figure 4B). The wound closure rate was also 48% lower for the HfFucCS + MO-I-1151_Low_ and 82% for the HfFucCS + MO-I-1151_High_ combinations compared to the treatment with HfFucCS alone (Figure 4B). These data are also well supported by the migration of cells using transwell chambers, where treatment with inhibitors showed reduction in cellular migration (Appendix A). The additive effect of the combination treatment indicates independent mechanisms of action and could be potentially explored therapeutically.

### 2.5. Combination of MO-I-1151 with HfFucCS Inhibitors Reduces the CAF-Mediated Invasion of Tumor Cells

The effect of the combination treatment of MO-I-1151 and HfFucCS was tested on the CAF 61137-mediated invasion of cancer cells in Matrigel-embedded spheroids. Compared to the untreated control, the combined treatment showed reduced invasion of spheroids (Figure 5A). HfFucCS reduced the area (20%, *p* = 0.14) and inverse circularity (52%, *p* < 0.0001) compared to the control (Figure 5B,C). The combined treatments reduced the area of the spheroids by 47% (*p* < 0.0001) for HfFucCS + MO-I-1151_Low_ and 48% (*p* < 0.0001) for HfFucCS + MO-I-1151_High_ (Figure 5B); inverse circularity was reduced by 68% (*p* < 0.0001) for both HfFucCS + MO-I-1151_Low_ and HfFucCS + MO-I-1151_High_ (Figure 5C). Thus, the combined treatment proved to be effective in blocking the invasion of the cancer cells supported by the CAF 61137.

## 3. Discussion

HNSCC is an assembly of epithelial squamous malignancies of the oral cavity, larynx, and pharyngeal regions and the seventh most common cancer worldwide [35]. Approximately 50,000 new cases and 12,000 cancer deaths occur annually in the US [36]. Smoking and alcohol are the dominant etiology of this mostly preventable disease; however, the etiology of up to 80% of oropharyngeal cancers is related to human papillomavirus (HPV) [37]. This is important because the tumors of HPV origin have different molecular pathogenesis and a better prognosis. Our study emphasizes analysis of the oral cancers (OSCCs) because they are most frequent and have common etiology, disease characteristics, and treatment approaches. Surgery is the standard treatment, but retained functionality and comorbid conditions affect selection of the therapeutic regimens [36]. Introduction of Cetuximab, inhibiting the epidermal growth factor receptor (EGFR) pathways activated in a majority of HNSCCs, and inhibition of immune checkpoints added new therapeutic options [38,39,40,41]. However, the survival of HNSCC patients remains largely unchanged [42,43,44], and further advances in molecular diagnostics and treatment at every stage of the disease are needed [45].

ASPH is an oncogenic protein upregulated in many types of cancer. Our study documents the upregulation of ASPH in HNSCC and shows that inhibition of ASPH activity by SMI MO-I-1151 limits HNSCC cancer cell migration and CAF-supported invasion into Matrigel.

Our results from the IHC analysis of patient tissues show that ASPH scores (intensity and distribution) are high in primary tumors of the oral cavity at all stages of the disease. The expression of ASPH in positive lymph nodes is as high as the expression in primary tumors (Figure 1), and it appears that even CIS lesions have already elevated ASPH compared to the adjacent normal tissue. The CIS representation on our tumor array is admittedly low, and this observation needs to be verified; however, overall, the upregulation of ASPH in the tumors is clearly demonstrated. The high expression of ASPH, even in early-stage tumors and CIS lesions, suggests that ASPH should be further examined as an early detection marker of HNSCC.

The uniformly high expression of ASPH prevented a meaningful study of the impact of ASPH on patient survival, reported in other malignancies [5,10,11]. However, the expression of ASPH in the primary tumors of HNSCC increases with the grade of the tumor. Poorly differentiated tumors express the most ASPH, and these tumors tend to be most aggressive. The intensity and distribution of ASPH in the stroma of the tumors is also elevated compared to the adjacent normal tissues. The distribution of the staining in the stromal cells is quite uniform, and the stromal cells contribute significantly to the content of ASPH in the HNSCC tissues. CAFs represent an important stromal cell type, and the CAF cells are clearly expressing ASPH, in line with prior single-cell RNA sequencing studies. Thus, the inhibitors of ASPH could be affecting the tumors by acting on both the tumor cells and on the CAF.

Given the high expression of ASPH in HNSCC tumors, we evaluated if ASPH inhibitors affect cancer cell migration and invasion in line with previous studies showing that ASPH regulates cellular migration, invasion, epithelial–mesenchymal transition, apoptosis, and stemness in other cancers [1,2,5,21]. We selected for our study MO-I-1151, a well-tested inhibitor of the β-hydroxylase activity of ASPH, with anti-tumor effects in other malignancies. MO-I-1151 reduced the invasion of colorectal cancer cells [22], reduced in vivo growth of cholangiocarcinoma tumor models [3], altered cellular senescence, and inhibited hepatocellular carcinoma growth and progression in pre-clinical in vivo models [1]. We tested the MO-I-1151 in scratch-migration assay, transwell chamber assay, and a spheroid co-culture cancer cell invasion model combining SCC35 cancer cells with a CAF (CAF 61137) derived from a primary head and neck tumor. The co-culture spheroids generated from these lines create a suitable invasive microenvironment that mimics the local microenvironment of HNSCC tumors, typically rich in stroma. MO-I-1151 clearly inhibits the CAF 61137-supported invasion of cancer cells into Matrigel in our co-culture spheroid model. The inhibitor significantly reduces the area of the invading spheroids and their inverse circularity, a measure of invasive extensions at the periphery of the spheroids. MO-I-1151 also showed a significant effect on inhibiting the migration of SCC35 cells over a period of 24 to 48 h in a wound closure assay and transwell migration chambers. This indicates that the ASPH inhibitor MO-I-1151 arrests the migration and invasion of SCC35 cells, whether they are in a monolayer culture or in a 3D spheroid model, where CAF 61137 stimulates their invasion into the extracellular matrix (ECM). The results suggest that inhibition of ASPH with MO-I-1151 could be an effective anti-invasive strategy and should be further tested using in vivo models.

We established in a recent study that HfFucCS, a marine fucosylated chondroitin sulfate isolated from the sea cucumber species *H. floridana*, previously studied as an anti-SARS-CoV-2 and anticoagulant agent [46], blocks the activity of heparan 6-O-endosulfatase SULF2 and inhibits the invasion of HNSCC cancer cells into the ECM [34]. The heparan 6-O-endosulfatases are upregulated in HNSCC tumors and are a prognostic marker of poor HNSCC outcomes [47,48,49]. SCC35 is an HNSCC-derived cancer cell line with high invasive potential and high expression of SULF2 [34,50]. SULF2 knockout in SCC35 significantly reduced the invasion of cells in vitro, which was also achieved by treating wild-type cells with HfFucCS [34]. Thus, in this study, we wanted to test the combination of the HfFucCS and MO-I-1151 inhibitors as an efficient strategy to block cancer cell invasion. We found that the combination of the two compounds has an additive effect on the CAF-supported spheroid invasion and on cellular migration assays. This is a first report of the effect of MO-I-1151 on inhibiting ASPH in HNSCC cancer cells and arresting cellular migration and invasion. Testing of the inhibitors in additional HNSCC cell lines, including HPV-positive cell lines, will be needed to confirm the results. In addition, we observe an additive effect when combining MO-I-1151 with HfFucCS, an effective inhibitor of HNSCC cell invasion. Further mechanistic studies are needed to determine how the inhibition of these enzymes in cancer cells and/or the CAF affects their communication and the invasion into Matrigel. It will also be important to evaluate how these cytostatic modulators of invasiveness affect tumor growth in combination with standard cytotoxic drugs and treatments as neo-adjuvant therapies.

## 4. Materials and Methods

### 4.1. Materials

Cell culture media (DMEM/F12, IMDM) and Growth Factor Reduced Matrigel were from Corning Inc., Corning, NY, USA. Hydrocortisone was from MilliporeSigma, Burlington, MA, USA. Cell culture supplements were from Gibco^TM^, Billings, MT, USA. Cell culture flasks and dishes were from Nunc, Thermo Fisher Scientific, Rochester, NY, USA; 96-well round-bottom low attachment plates were from Corning, Corning, NY, USA. Monoclonal anti-ASPH-antibody was from Santa Cruz (SC271391). DAB chromagen was from Agilent (Dako #K3468), and Hematoxylin was from Thermo Fischer, Memphis, TN, USA (Harris Modified Hematoxylin).

### 4.2. Study Subjects and Samples

Participants were enrolled between 1995 and 2020 in collaboration with clinicians at the Department of Otolaryngology—Head and Neck Surgery at Georgetown University Hospital under a protocol approved by the Georgetown University Institutional Review Board. Patients with newly diagnosed HNSCC primarily from the oral cavity (n = 155) undergoing surgical resection were selected for our study. HNSCC diagnosis was based on medical examination and was confirmed by histopathological evaluation of the tissues. Cancer classification was based on the 7th Edition of the American Joint Committee on Cancer Staging manual. An overview of the demographic and clinical features of the cohort is described in Table 1. The majority of the patients had early stage tumors (stage 1 and 2), with 36% (T1), 33% (T2), 8% (T3), and 22% (T4), and 48% of the patients had positive lymph nodes (LN+) with an N-stage distribution of N1 (n = 28), N2 (n = 42), and N3 (n = 4). The tumor grades of the patients were classified as well- (w = 10%), moderately (m = 59%), and poorly (*p* = 31%) differentiated. Disease recurrence was reported in 41% of the patients. The median follow-up time of the patients was 42 months (range: 0.2 to 286 months), with the outcomes of alive (n = 97), deceased (n = 56), and unknown (n = 2). The population distribution of patients reflects the demographics seen at the Georgetown University Hospital, with median age of the population of 62 years (range: 24–93) years.

Gene expression and proteomic data for ASPH were analyzed in the TCGA (Cancer Genome Atlas Consortium) and CPTAC (Clinical Proteomic Tumor Analysis Consortium) HNSCC datasets as described previously [48,49]. Briefly, 43 HNSCC patients with RNA-seq data available for both tumor and adjacent benign tissues were analyzed for ASPH differential expression, and log-transformed fold-changes (log_2_FC) were computed by difference of log_2_(counts+1) of RNA-seq data. The proteomic data and clinical information of 64 HNSCC patients, including matched tumor and adjacent non-tumor tissue pairs, were obtained from CPTAC. The quantification of protein abundance was conducted using the CPTAC Common Data Analysis Pipeline, which determined the log2 ratio of individual proteins to an internal control using only peptides not shared between quantified proteins.

### 4.3. Tissue Microarray and Histological Sections

Tissue microarray and histological sections were generated by the Lombardi Comprehensive Cancer Center’s Histopathology and Tissue Shared Resource, Georgetown University Medical Center, using a T-sue Microarray mold kit from Electron Microscopy Sciences [51]. Briefly, tissue cores were taken from paraffin-embedded blocks using a 1.5 mm punch tip from Electron Microscopy Sciences and manually inserted into the recipient paraffin blocks. Sections of 5 µm were cut from the array using a Leica tissue microtome and analyzed by immunohistochemistry (IHC). We evaluated 155 patient cases spread across 6 TMAs in 314 cores. The array distribution scored for ASPH expression was as follows: Primary lesion (92 cases, 144 cores); Primary + LN (15 cases, 46 cores); Primary + Adjacent normal (31 cases, 90 cores); Primary + Normal + LN (2 cases, 8 cores); Primary + CIS (Carcinoma in situ) (2 cases, 5 cores); CIS (1 case, 1 core); LN (10 cases, 18 cores); Normal tissue (2 cases, 2 cores).

### 4.4. Immunohistochemistry and Pathology Scoring

TMAs of patient tissue samples were stained using anti-ASPH monoclonal antibody (1/100 dilution) and the DAB chromogen kit following the manufacturer’s instructions. Slides were counterstained with Hematoxylin (Fisher, Harris Modified Hematoxylin), blued in 1% ammonium hydroxide, dehydrated, and mounted with Acrymount. Consecutive sections with the primary antibody omitted were used as negative controls. All TMAs were scanned with the Aperio image scope (Aperio GT450) and scored by a pathologist. Staining intensities were classified as 0  =  negative, 1  =  weak, 2  =  moderate, and 3  =  strong. The proportion of positive tumor epithelial cells was assessed, and the staining distribution was classified as 0  =  no cells, 1  =  1–25% of cells, 2  =  26–50% of cells, and 3  > 50% of cells. Positive fibroblast cells in the stromal compartments were evaluated separately using the same scoring criteria.

### 4.5. Cell Culture

The SCC35 cell line, a human squamous cell carcinoma cell line derived from a hypopharyngeal tumor, was kindly provided by Prof. Vicente Notario, Georgetown University. Primary head and neck fibroblast CAF 61137 was derived at the Princess Margaret Cancer Centre, Toronto, CA, from a post-surgery tumor sample of a patient (male/59 years of age) before radio/chemotherapy, in line with an approved Research Ethics Board protocol. The tumor site was the tongue, and the tumor was categorized as SCC, stage IVA, T3/N2b/M0. SCC35 cells were grown in DMEM/F12 supplemented with 400 ng/mL hydrocortisone. CAF 61137 cells were grown in IMDM. The cells used in this study were in passage <10 and in log-phase growth. All media were supplemented with 10% fetal bovine serum, non-essential amino acids, and 1 mM sodium pyruvate and were grown in a humidified 5% CO_2_ atmosphere. Cells were subcultured at a 1:3 ratio at 90 to 100% confluence.

### 4.6. Wound Healing

SCC35 cells were cultured in 96-well plates (1 × 10^4^ cells/well) to form a confluent monolayer. After 24 h of incubation, scratches were created by scraping the monolayer cells using a 96-well pin block in Woundmaker^TM^ (Essen Bioscience, Sartorius, Göttingen, Germany). Subsequently, the cells were gently washed with serum-free medium to remove dislodged cells. The cells were treated with two doses of MO-1151 (5 μM, 25 μM) to inhibit invasion either alone or in combination with HfFucCS (10 µg/mL) and observed at defined time intervals (0 h, 24 h and 48 h) after scraping. The migration of cells was analyzed by the area of the wound and quantified as follows: percentage wound closure = (area of initial wound at time t0—area of wound at time t1)/area of initial wound at time t0 × 100.

### 4.7. Spheroid Invasion

Spheroids were generated as described previously [34]. For co-culture spheroids, an equal number of SCC35 and CAF 61137 cells (3 × 10^4^ cells/mL for each cell type) were mixed and seeded in ultra-low attachment 96-well round bottom plates, and the cells were grown for 1 day in an incubator followed by Matrigel embedding. Spheroids were cultured in an incubator and imaged using an Olympus IX71 inverted microscope (Olympus, Tokyo, Japan). Digital images of spheroids were analyzed using Particle Analysis in ImageJ software version 1.51p (NIH, Bethesda, MD, USA) to obtain the values of area and inverse circularity. The area for spheroids is the count of pixels comprising the object. Inverse Circularity = $\frac{1}{Circularity}$, where circularity (*Circularity* = 4*π* × [*area*]/[*perimeter*]) is a property of the spheroid calculated using automated image analysis with the analyze particles tool. A circularity value of 1 (maximum) arbitrary unit (A.U.) indicates that the spheroid is perfectly circular; decreasing values indicates deviations from a circular spheroid. All our data processing and analysis parameters are identical for samples in the same dataset.

### 4.8. Inhibitor Treatment of Spheroids

Co-culture spheroids were treated with two doses of MO-I-1151 ASPH inhibitor (5 μM, 25 μM) with/without HfFucCS (10 µg/mL) and compared to non-treated spheroids (NTC). The inhibitors were added to the complete medium prior to treatment, and the solutions were gently added to the Matrigel spheroids. The plates were incubated for 5 days and imaged on the day of harvest using a phase-contrast microscope (Olympus). The results were quantified using Image J analysis version 1.51p (NIH, Bethesda, MD, USA), as described above.

### 4.9. Statistical Analysis

GraphPad Prism version 10 for Windows (GraphPad Software, La Jolla, CA, USA) was used for statistical analysis, with summary outcomes represented as mean values ± SD. Differential expression between the tumor and paired normal tissues was assessed through a paired *t*-test. IHC results were compared among normal, tumor, and lymph node samples using pairwise *t*-tests. Wound closure, area, and inverse circularity of the spheroids at day 5 were evaluated using one-way analysis of variance (ANOVA) with post hoc Tukey’s test. A two-sided *p* value < 0.05 was considered statistically significant unless specified otherwise.

## 5. Conclusions

Our study documents the upregulation of Aspartate β-Hydroxylase (ASPH) in head and neck squamous cell carcinoma (HNSCC) based on immunohistochemical examination of human tumor tissues. At the same time, we document the inhibition of cancer cell invasion in a spheroid co-culture model with cancer-associated fibroblasts (CAFs). We used the model to establish the ability of the ASPH inhibitor MO-I-1151 to inhibit the invasion of HNSCC tumor cells into Matrigel, and we showed an additive inhibitory effect of its combination with a recently described heparan 6-O-endosulfatase (SULF2) inhibitor, HfFucCS.

## Figures and Tables

**Figure 1 ijms-25-04998-f001:**
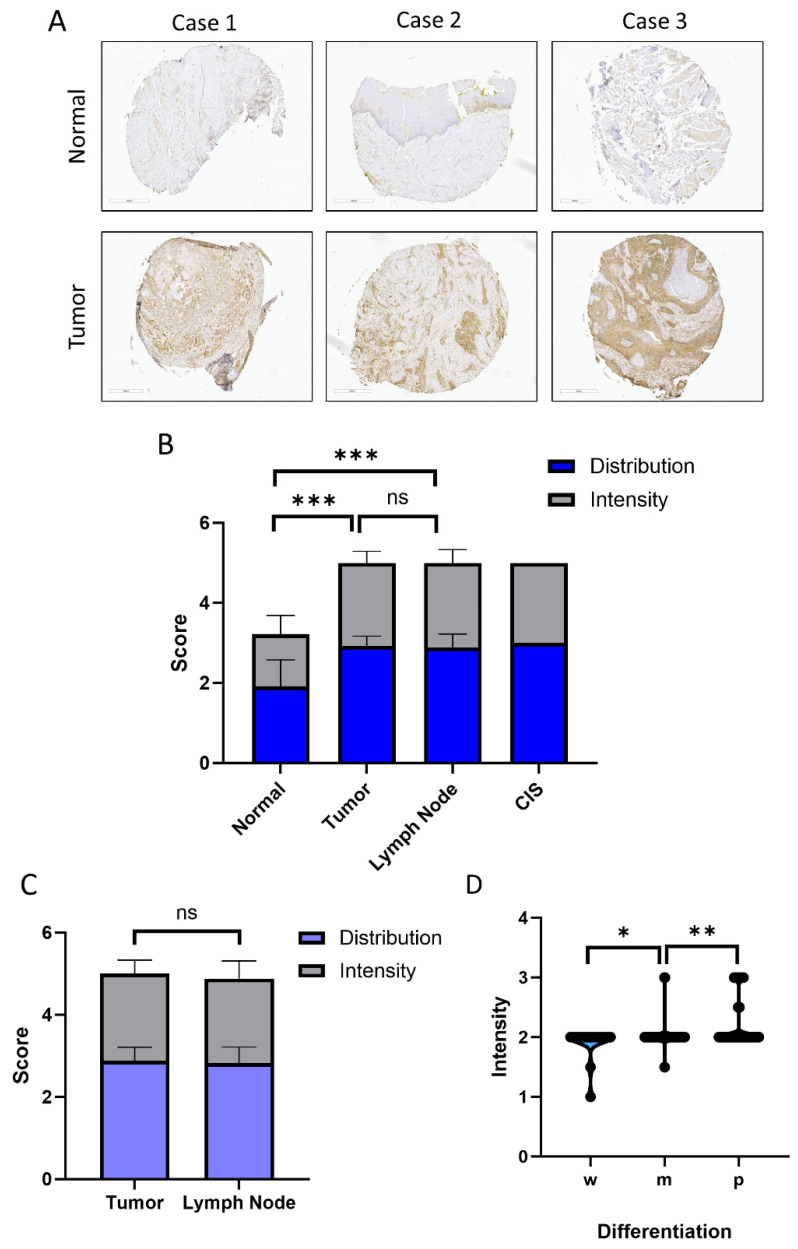
ASPH protein is elevated in the HNSCC tumor tissues. (**A**). Typical IHC staining for ASPH in HNSCC tumors and adjacent normal tissues. (**B**). ASPH staining scores (distribution and intensity) increase significantly from normal adjacent tissues (n = 35) to primary tumors (n = 142) and SCC-positive lymph nodes (n = 10). CIS scores appear equally high as tumors but low numbers of the cores (n = 3) prevent statistically powered evaluation. (**C**). ASPH score of paired tumors and SCC-positive lymph nodes (n = 17) show equally high expression of ASPH. (**D**). ASPH staining intensity in well- (w, n = 15), moderately (m, n = 92), and poorly (*p*, n = 48) differentiated tumors significantly increases. Statistical significance * *p* value < 0.05, ** *p* value < 0.001, *** *p* value < 0.0001, ns = not significant.

**Figure 2 ijms-25-04998-f002:**
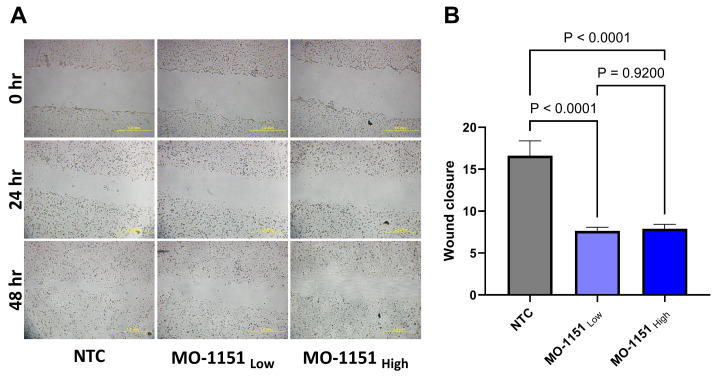
ASPH inhibitor MO-1151 arrests migration of SCC35 cells. (**A**). Representative images of a wound healing assay showing migration of cell at three timepoints (0, 24, 48 h) in non-treated control (NTC) and MO-1151 (low dose—5µM, high dose—25 µM) treated cells. Scale bar: 1 mm. (**B**). MO-1151 treatment shows significant difference in wound closure in the low and high doses compared to the control cells. Graphs represent mean value of each group with SD, n = 5 independent repeats. *p*-values for group-wise comparison are presented in panel (**B**).

**Figure 3 ijms-25-04998-f003:**
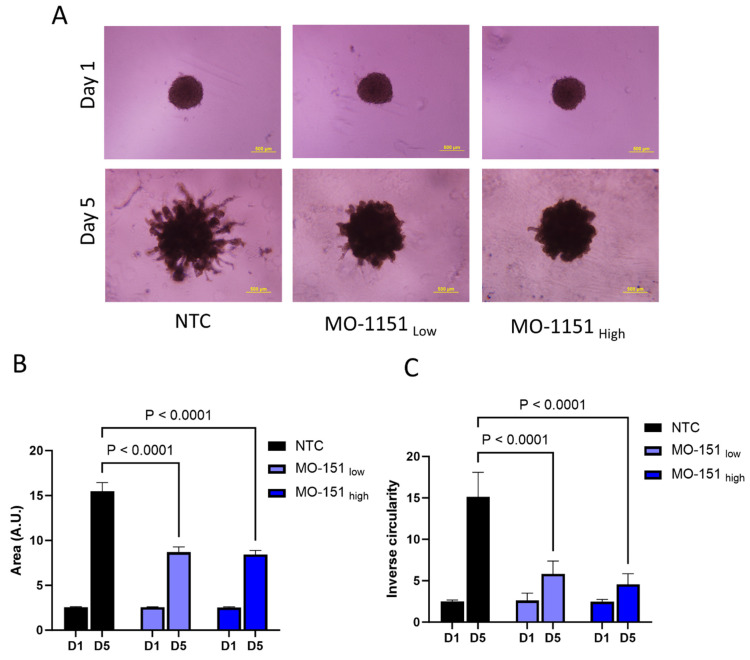
ASPH inhibitor MO-1151 reduces CAF-mediated invasion of SCC35 cells. (**A**). Representative images of a Matrigel invasion assay using co-culture spheroids (CAF 61137 + SCC35 cells) showing reduced cellular protrusions in MO-1151-treated spheroids (Low dose—5µM, high dose—25 µM) compared to NTC on day 5. Scale bar: 500um. (**B**). MO-1151 treatment leads to significantly reduced spheroid area on day 5 compared to the NTC. (**C**). MO-1151 treatment leads to significantly reduced inverse circularity of the spheroids on day 5 compared to the NTC. Graphs represent mean value of each group with SD, n = 5 independent repeats. *p*-values for group-wise comparison are presented in panels (**B**,**C**).

**Figure 4 ijms-25-04998-f004:**
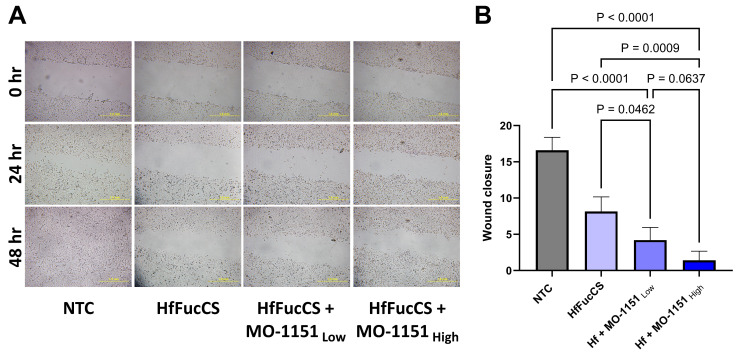
Treatment of SCC35 cells with a combination of the MO-1151 and HfFucCS inhibitors shows additive effect on migratory arrest. (**A**). Representative images of a wound healing assay showing migration of SCC35 cells at three timepoints (0, 24, 48 h) in non-treated control (NTC), HfFucCS (10 μg/mL), and HfFucCS (10 μg/mL) with MO-1151 (Low dose—5 µM, and high dose—25 µM) treated cells. Scale bar: 1 mm (**B**). Combination treatments significantly reduced the wound closure compared to the NTC or HfFucCS treatment alone. Graphs represent mean value of each group with SD, n = 5 independent repeats. *p*-values for group-wise comparisons are presented in panel (**B**).

**Figure 5 ijms-25-04998-f005:**
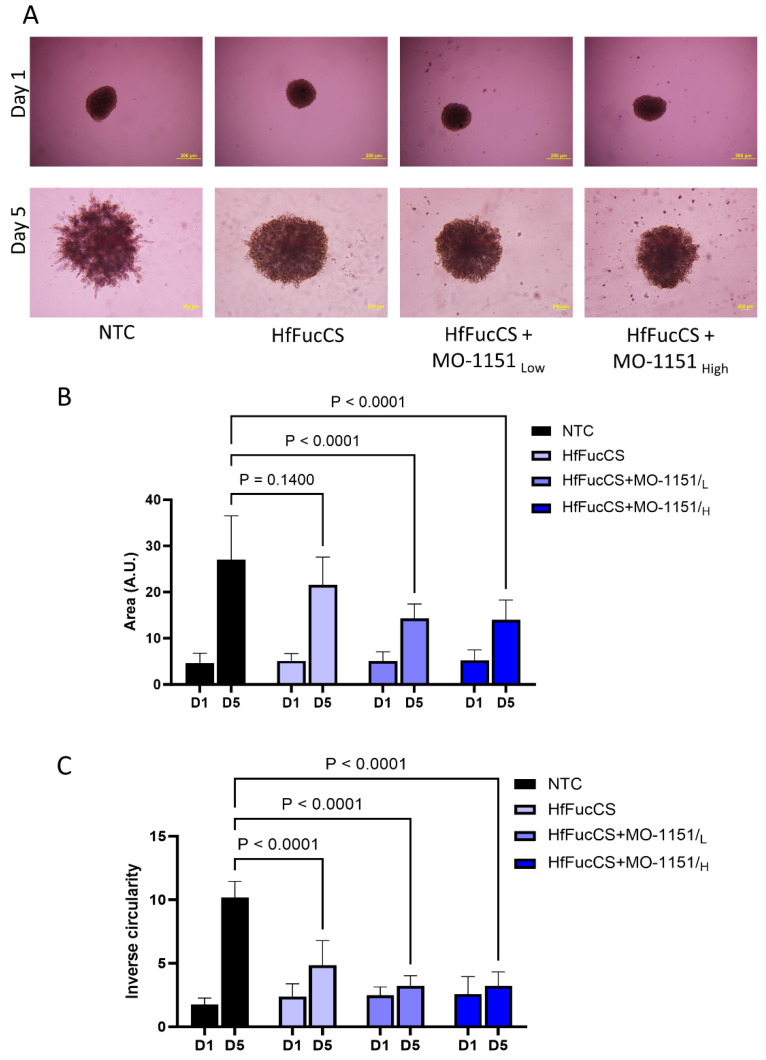
MO-1151 and HfFucCS additively reduce invasion of the co-culture spheroids into the Matrigel. (**A**). Representative images of Matrigel invasion of co-culture spheroids (CAF 61137 + SCC35) show reduced cellular protrusions in spheroids treated with HfFucCS (10 mg/L) or HfFucCS (10 mg/L) + MO-1151 (Low dose—5 µM, high dose—25 µM) compared to non-treated control (NTC) on day 5. Scale bar: 200 µm. (**B**). MO-1151 + HfFucCS combined treatments significantly reduced spheroid area or (**C**). inverse circularity on day 5 compared to NTC. Graphs represent mean value of each group with SD, n = 5 independent repeats. *p*-values for group-wise comparison are presented in panels (**B**,**C**).

**Table 1 ijms-25-04998-t001:** Demographic and clinicopathological characteristics of the study population.

Variable	Parameter	No. of Cases	Percentage %
Gender	Male	92	59
	Female	63	41
Age	<60 y	66	43
	≥60 y	86	56
	Unknown	3	2
Race	CA	115	74
	AA	11	7
	Asian	2	1
	Other	11	7
	Unknown	16	10
T-Stage	Early (T1, T2)	107	69
	Late (T3, T4)	46	30
	Unknown	2	1
Node	N+	74	48
	N-	79	51
	Unknown	2	1
M-Stage	M+	1	1
	M-	152	98
	Unknown	2	1
Disease Recurrence	Yes	64	41
	No	72	47
	Unknown	19	12
Margin	Positive	18	11
	Negative	69	45
	Close	58	37
	Unknown	10	7
Grade	w	15	10
	m	92	59
	*p*	48	31

## Data Availability

The data presented in this study are available in this article.

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
