# Peer review of "Aspartate β-Hydroxylase Is Upregulated in Head and Neck Squamous Cell Carcinoma and Regulates Invasiveness in Cancer Cell Models"

_ijms, 2024, doi:10.3390/ijms25094998_

Round 1
Reviewer 1 Report
Comments and Suggestions for Authors
The authors put commendable effort into pointing out ASPH as a probable target but offer little insight into the mode of action. However, the importance of ASPH in solid and metastatic tumors needs to be laid out better, and the relevance to the usage of HNSCC needs to be clarified. Further, HNSCC is not highly metastatic cancer as compared to breast or prostate cancers. Inhibition studies would be better justified on highly metastatic cancers. The addition of a metastatic in vivo cancer model to test ASPH would boost the significance.
The in vitro data presented in this study is compelling and warrants further exploration in in vivo OSCC models. However, the assays were limited to spheroid migration and invasion, providing no mechanistic insights. Future research could focus on assessing the cellular and molecular pathways that underlie these migration and invasion properties, thereby enhancing our understanding of ASPH's role in oral cancers.
What is the inhibitors' ED50? The rationale for using a single low and high dose must also be included.
In vitro assays were conducted only on a single cell line. Four to five candidate cell lines with varying HPV status would be ideal.
Author Response
We thank the reviewer for the valuable comments.

Reviewer 2 Report
Comments and Suggestions for Authors
Reviewer report
Aspartate β-Hydroxylase is upregulated in head and neck squamous cell carcinoma and regulates invasiveness in cancer cell models.
This is the well-designed study showing the upregulated levels of Aspartate β-hydroxylase (ASPH) in case of various stages at HNSSC. Similarly, the study shows use of two different inhibitors MO-I-1151 with HfFucCS as small molecule inhibitors in SCC-35 in-vitro.
1. Please explain the entire study in the form of graphical abstract and flow chart which will be easy to understand.
2. What was the exclusion and inclusion criterion of the study please explain.
3. Will the outcome of the study remain same in both male and female patients irrespective of gender difference please explain.
4. What is the rationale of selecting SCC35 for the study please justify.
5. What was the passage number used in the current study please specify.
6. Which standard drug was used in the study to compare the activity please specify.

Author Response

(The authors gave the same response as above.)

Round 2
Reviewer 1 Report
Comments and Suggestions for Authors
The authors put commendable effort into pointing out ASPH as a probable target but offer little insight into the mode of action. Though authors have been moderately responsive to the previous critique, the in vitro data presented in this study is compelling. And it is recommended that authors continue to explore ASPH in in vivo OSCC models.